# Determination of Interlaminar Shear Properties of Fibre-Reinforced Composites under Biaxial Loading: A New Experimental Approach

**DOI:** 10.3390/polym14132575

**Published:** 2022-06-24

**Authors:** Kirill Guseinov, Oleg Kudryavtsev, Alexander Bezmelnitsyn, Sergei Sapozhnikov

**Affiliations:** Department of Engineering Mechanics, South Ural State University, 454080 Chelyabinsk, Russia; kudriavtcevoa@susu.ru (O.K.); bezmelnitsynav@susu.ru (A.B.); sapozhnikovsb@susu.ru (S.S.)

**Keywords:** polymer matrix composites (PMCs), biaxial loading, shear strength, nonlinear behaviour, thick-walled composites, failure criteria, woven carbon/epoxy laminates

## Abstract

The complexity of biaxial tests and analysis of their results makes it difficult to study the interlaminar shear properties of fibre-reinforced composites, particularly under through-thickness compression, which occurs in thick-walled composite elements. The improvements in experimental methods to study the features of the nonlinear behaviour of composites under biaxial loading is now an important and relevant task in the development aircraft structural elements made of carbon fibre-reinforced polymers. This study aimed to develop a new experimental approach for the reliable determination of the interlaminar shear properties of laminates under through-thickness compression using a standard testing machine. An appropriate V-notched specimen was developed based on the configuration of well-known Iosipescu and butterfly-shaped specimens. The approach is demonstrated using woven carbon/epoxy laminates. Both the preliminary assessment of the stress fields under combined compression/shear loading and the analysis of fracture mechanisms were performed with finite-element modelling in a three-dimensional formulation. The digital image correlation (DIC) method was used to obtain experimental, full-field deformations of the specimens and to estimate the uniformity of the strain distribution in the gauge section. The stress–strain curves were obtained under biaxial loading, and the corresponding features of the composite failure behaviour were analysed in detail. It was found that the maximum compression strain on the stress–strain curves, in some cases, corresponded to the discontinuity in the composite structure. In these cases, the disproportionate changes in through-thickness strains in the gauge section of the specimens were recorded at the maximum load. With the increase in through-thickness compression stresses, the difference between the shear strength values, determined by the maximum load and the maximum compressive strain, increased by up to 20%. It was shown that the assessment of the composite strength at maximum load at the design stage significantly increased the risk of premature failure of the composite elements during exploitation.

## 1. Introduction

Carbon-fibre-reinforced polymers (CFRP) are considered as an alternative to metals in various structural applications where weight is one of the critical design parameters. These materials are widely used in modern high-performance structures due to advantages such as high specific strength, stiffness, and good fatigue resistance [1,2]. For aircraft engine structures, the use of CFRP for fan blades and protective casings of turbofan jet engines is of special interest. The use of CFRP helps to reduce the mass not only of the elements themselves but also of the assembling parts due to the reduction in the transmitted inertial and dynamic loads. One of the current trends in the development of aircraft structural elements from polymer matrix composites (PMCs) is the replacement of expensive and time-consuming tests of real prototypes with virtual tests based on high-precision numerical models. Thus, it is necessary to use reliable models of materials verified by the experimental results. The mechanical behaviour of a composite material under the multiaxial loading that occurs during the operation of thick-walled elements, including nonlinear deformation caused by the progressive damage accumulation due to matrix cracking, delamination, and fibre fracture, is of great importance [3,4,5,6]. The interlaminar shear strength of the composite determines the load-bearing capacity of thick-walled composite elements in most cases [7]. At the same time, the experimental data indicated that through-thickness compression could significantly increase the interlaminar shear strength of the laminates [8]. Not considering this fact can lead to decreases in the weight efficiency of the composite structure [9].

To implement hidden strength reserves in the material, it is necessary to correctly select the failure criterion. To evaluate PMCs strength under biaxial loading, various failure criteria, such as Northwestern University theory (NU-Daniel) [10,11], Christensen et al. [12], Puck IFF [13], and Mohr–Coulomb [14,15] have been proposed. In [16,17], several failure criteria presented in the literature were compared with the experimental results. Although all failure theories explained the trend towards increasing shear strength under through-thickness compression, and were in good agreement with the experimental results, criteria such as Mohr–Coulomb and the Puck inter-fibre fracture (IFF) needed to define additional parameters, for example, using biaxial tests. The NU-Daniel theory and Christensen criteria do not require a curve-fitting parameter, since these failure criteria are expressed in terms of the material’s strength properties. In [17], it was concluded that the Daniel criterion is preferable, since it is exclusively described by the properties of the material, without additional parameters. It should be noted that many existing failure criteria of laminates are consistent with various experimental data. One of the reasons for this uncertainty is the lack of adequate and reliable experimental data, particularly in biaxial shear tests.

At present, there are several methods for the experimental determination of the properties of laminates under quasi-static biaxial loading [8,17,18,19,20,21,22,23]. To allow for a detailed evaluation of the effectiveness of the different approaches, researchers [19,20,21,22] used 2D digital image correlation (DIC) to obtain full-field strain measurements, thus identifying damage during loading. In [24,25], a comprehensive review of the digital image correlation technique is presented. Gan et al. [17] modified the double-notch shear test to measure the interlaminar shear strength of unidirectional CFRPs under moderate through-thickness compression. The tests were carried out on a biaxial testing machine equipped with four independent hydraulic grips. The interlaminar shear strength under moderate through-thickness compression increased by ~2 times. This experimental study aimed to determine the best failure criterion to evaluate the interlaminar strength of the composite under combined loading. The researchers experimentally confirmed the increase in interlaminar shear strength without analysing the mechanical behaviour of the material. DeTeresa et al. [8] studied the effects of through-thickness compression on the interlaminar response of five different composites reinforced with glass and carbon fibres. Hollow cylindrical specimens were preliminarily compressed in the axial direction, and then were subjected to torsion, thus implementing a combination of through-thickness compressive and interlaminar shear stresses. Their results showed that the maximum increase in shear strength was material-dependent and varied from 55% for the E-glass fabric composite to 340% for the laminated T300/F584 composite. At the same time, the failure curves indicated that, for the increase in shear strength up to the maximum value, the behaviour was quadratic. However, it was not identified whether the shear failure was due to interlaminar stresses, intralaminar stresses, or a combination of the two. The cruciform specimens were often chosen to obtain the in-plane mechanical performance of composite laminates under biaxial loading. The successful testing of cruciform specimens requires softening the stress in the regions surrounding the biaxially loaded zone (i.e., arms and corners), or increasing the stress produced in the central region by reducing the thickness [18]. Furthermore, the implementation of the tests required complex and expensive test equipment. Koerber et al. [19] used an alternative approach based on the transverse compression of cubic unidirectional CFRP specimens that were cut at different angles to the loading direction. An average increase of 25 and 42% was observed for the in-plane shear modulus and shear strength for the combined transverse compression and in-plane shear loading. However, the researchers pointed out that friction also affected the quasi-static response; therefore, the stress–strain curves no longer reflected the actual behaviour of the material under combined loading. Cui et al. [20] also noted that the concentration at the specimen ends, caused by friction, prevented a uniform stress distribution over the cross-section and affected the obtained results. Gan et al. [21] used a modified Arcan fixture (MAF) to study the mechanical response and strength of unidirectional GFRP with different stacking sequences under combined in-plane loading. This fixture allows for combined tension/compression-shear loading to be applied to the specimen. The authors demonstrated that the properties of the GFRP under combined loading could be obtained with high accuracy using the considered experimental approach. However, the authors noted that the design of the fixture and specimen geometry should be modified to study the mechanical behaviour of CFRPs. In [22,23], the modified test fixture was equipped with guide units that increased the stiffness of the apparatus. However, the researchers analysed the behaviour of CFRPs for combined transverse compression and in-plane shear loading, while the possibility of using a new device to determine interlaminar shear properties under the through-thickness compression loading was not considered.

Clearly, there are no generally accepted test methods for reliably determining the interlaminar shear properties of composites due to the complexity of biaxial tests and the difficulties in interpreting their results. At present, there is a need to develop alternative experimental methods to study the effect of through-thickness compression on interlaminar mechanical behaviour and the strength of composites under biaxial loading using widespread test equipment. It should also be noted that most of the studies were devoted to studying unidirectional composites, while the nonlinear behaviour of fabric composites under biaxial loading were not studied as often.

This study aimed to develop a new experimental approach to obtain valuable biaxial experimental data, such as interlaminar shear strengths and shear moduli, and shear stress–strain response, which can be used to develop and validate nonlinear numerical material models. The features of the nonlinear behaviour of woven CFRP, which is widely used in the manufacture of fan blades for jet engines and composite elements with bolted joints [16,26], was studied. A new fixture and V-notched specimen were developed for composite testing. The new fixture provides biaxial loading of specimens using widespread test equipment. The shape and dimension of the specimen were selected to ensure stable fixation and uniform strain distribution in the gauge section under different loading conditions. The digital image correlation (DIC) method was used to obtain the full-field deformations of the specimen and identify the failure mode. A uniform distribution of the shear strains in the gauge section of the specimen is achieved when the new fixture is used for testing. Then, stress–strain curves were obtained under combined compression–shear loading, and the main features of the woven CFRP deformation were considered. It was found that the maximum compression strain on the stress–strain curve corresponded to the moment at which the composite structure discontinuity occurs. Finally, the experimentally obtained values of interlaminar shear strength under combined loading were compared with NU-Daniel theory to confirm their reliability. An increase in interlaminar shear strength by 1.3–2.25 times was observed at the loading angles α = 20–45°. The difference between the shear strength values, determined by the maximum load and the maximum compressive strain, can reach 20% under significant through-thickness compression loading.

## 2. Materials and Methods

### 2.1. Experimental Setup and Design of the Specimen

Figure 1 shows the fixture that was developed to determine the mechanical properties of PMCs under combined compression–shear loading. All parts of the test fixture were made of high-carbon steel. Two plates resembling a 40-mm disc were used to fix the specimens. The appropriate size was chosen based on the maximum load limit of the test equipment (100 kN). Grooves were made in the plates to provide certain fastening forces for a specially shaped specimen. The specimen was glued to the grooves, and additional steel tabs were used for its precise position and strong fixation. The load on the plates was transferred through two supports with cylindrical surfaces of the same radius as the plates. This provided a smooth change in the through-thickness and interlaminar shear stress ratio in the specimen. The displacement of the lower support in one plane was implemented using several steel rollers.

Well-known Iosipescu and butterfly-shaped specimens [27,28] guided the choice of the specimen configuration for testing in the new fixture. This specimen configuration makes it possible to implement nearly uniform shear-stress distribution in the gauge section between the notches. The modifications to the V-notched specimen and the loading scheme are shown in Figure 2. The fibre directions in the woven composite are labelled as follows: 1 is warp direction, 2 is weft direction, and 3 is out-of-plane direction.

The desired combination of interlaminar shear and through-thickness compression stresses in the gauge section of the specimen was obtained by varying the loading angle α (Figure 2b). Through-thickness compression and interlaminar shear stresses in the gauge section of the specimen are defined as follows.
(1)$$\sigma_{3}=\frac{P\cdot \cos(\alpha )}{l\cdot t},$$
(2)$$\tau_{13}=\frac{P\cdot \sin(\alpha )}{l\cdot t}.$$ Here, *P* is the applied load, *l* is the length of the gauge section, *t* is the specimen thickness, σ_3_ is the through-thickness compression stresses, τ_13_ is the interlaminar shear stresses.

A common problem in the biaxial composite test is specimen failure from the specimen due to the coupling forces created by the connecting bolts [20]. For the proposed experimental setup, there was a similar risk of premature specimen failure with a significant compressive load from the fastening forces. In this regard, it was necessary to find the specimen dimensions, not only to ensure a uniform stress distribution in the gauge section, but also to prevent early specimen fracture due to contact stresses.

Parameters such as notch angle, depth and specimen width have little effect on stress uniformity [29]. Therefore, the 90° notch angle and 4-mm notch depth suggested in ASTM D5379 [27] were chosen for the modified specimens in this study. The specimen width of 10 mm was chosen to prevent local buckling of the composite.

One of the most common problems is related to the premature failure of V-notched specimens due to the stress concentration at the notch root. The stress concentration at the notch root with a radius of 1.27 mm had a direct effect on the experimental results [29]. In [21,30], the researchers used different radii at the notch root when performing biaxial static tests on the butterfly-shaped specimen. Gan et al. [21] chose a notch radius of 5 mm, guided by the technological limitations of waterjet cutting. Gning et al. [30] used the notch radius of 2.5 mm, which provided a more uniform shear-strain distribution in the gauge section. At the same time, the authors noted that the decrease in the notch radius made it possible to obtain a more uniform shear-stress distribution in the gauge section. For this reason, the smallest technologically accessible notch radius of 0.9 mm was chosen for the proposed specimen.

Gauge length plays an important role in ensuring the uniformity of the stress distribution during biaxial composite tests. Gan et al. [21] and Hao et al. [22] used the gauge lengths of 28.56 mm and 25 mm for 1–2 plane biaxial tests of unidirectional GFRP with different stacking sequences and woven CFRP specimens, respectively. However, according to the data of The World-Wide Failure Exercise (WWFE-I, II and III) [31,32], failure under biaxial loading in the 1–3 plane occurred at much higher loads. It was decided to limit the gauge length between the notches to 17.3 mm to prevent premature failure of the specimen outside the gauge zone due to the fastening forces. In [30], a close value of the gauge length between the notches was used to study the interlaminar shear properties of GFRP.

Another important specimen parameter is height. On the one hand, the increase in the specimen height makes it possible to provide more reliable fastening and reduce the probability of preliminary failure from crushing. However, the influence of the specimen height on the stress distribution uniformity in the gauge section requires more in-depth study. To reduce the risk of premature failure of the specimen from the fastening forces created by the plates under high compressive loads, 20-mm height specimens were used. To assess the influence of the specimen height on the uniformity of the stress distribution in the gauge section and to determine the possible failure area under different compression-shear loading, finite-element simulations of the CFRP biaxial tests were carried out.

### 2.2. Materials and Specimens

The composite plates were made by hand lay-up with press moulding using a mixture of epoxy resin, diethylene glycol, and triethylenetetramine ED-20/DEG/TETA (82:8:10 by weight). A similar manufacturing process was used in [33]. The composites were reinforced with UWB-200-3K-Twill2/2-100 (3K, 200 g/m^2^) twill carbon fabric (UMATEX^®^, Moscow, Russia).

The initial curing of the 32-ply laminates with 0° lay-up was carried out in the press at 25 °C for 24 h under 2 MPa pressure. The cured laminates were bonded together to achieve a 22-mm total panel thickness. The post-curing was performed at 80 °C for 5 h. The fibre volume fraction, evaluated with the conventional burning-test method, was approximately 59%. The specimens for biaxial loading tests were cut from the composite plate by CNC-router. The cutting scheme and dimensions of the specimens for biaxial loading are shown in Figure 3.

### 2.3. Numerical Model

Many researchers [34,35,36,37,38] used detailed finite-element (FE) models reflecting the effect of through-thickness compression on composite interlaminar properties to predict the nonlinear behaviour of the material under biaxial loading. Experimental data on the non-linear deformation and failure of composites under biaxial loading were used to verify the parameters and calibrate the material models. However, there was no consensus among the authors on the preferred experimental procedure to obtain these data. Moreover, significant detailing and, as a result, the complexity of preparing such FE models, were only justified when studying the effect of various damage accumulation mechanisms on the composite nonlinear response under biaxial loading. In this study, numerical simulations primarily aimed to obtain a first approximation of the specimen stress fields under biaxial loading. In this regard, a linear elastic analysis was performed, and the nonlinear behaviour of composites in shear was not considered. It was assumed that the uniformity of the stress and strain fields will remain with an increase in loads beyond the limits of proportionality. Similar simplifications were previously used in [39] when studying the stress–strain state of Iosipescu specimens. This approach made it possible to assess the uniformity of the stress distribution in the gauge section for the selected specimen sizes. The numerical solution was performed using the finite-element method in ANSYS Workbench with the implicit time integration scheme.

An orthotropic linear elastic model was used to simulate mechanical behaviour. The typical elastic and strength properties of woven CFRP [26] were used in the simulation (Table 1). The interlaminar shear properties were additionally refined based on the results [33]. The authors demonstrated that the weave structure of the material was one of the factors that determined the interlaminar shear properties of woven composites. In this regard, the interlaminar shear strength of 54.3 MPa and the interlaminar shear modulus 3.4 GPa corresponding to twill-weave fabric CFRP were specified.

SOLID186, a high-order, volumetric, finite element, was used. The FE mesh was refined between the notches. The 0.25-mm FE size in the gauge section of the specimen was chosen by a mesh convergence study, similarly to [9]. The total number of finite elements was 92,480. A linearly elastic material model with steel properties E = 200 GPa, µ = 0.3 was used for plates, tabs and supports.

Several different contact formulations were used to obtain adequate specimen deformation during loading in the new fixture. Frictional contact type was implemented between the plates and supports in Augmented Lagrange formulation. The friction coefficient in the contact pair was set to 0.2. For the remaining contact pairs, it was decided to use the No separation contact type. This contact type made it possible to ensure the transfer of loads along the normal to the surface of the specimen, which led to some sliding of the plates, which occurred during biaxial loading.

Due to their symmetry, only half of the specimen was considered, and corresponding boundary conditions were applied. A load was applied to the upper support while the lower support had a degree of freedom for displacement in the direction perpendicular to the loading axis. Figure 4 shows the finite-element mesh, boundary conditions, and loads. The loading angle α varied with a step of 5°. According to NU-Daniel theory, the load was increased for each loading angle until failure was reached [10,11]. Strength assessment was carried out by an analysis of stress fields in the layer coordinate system. The strength was determined by the criterion of maximum stresses in the region of through-thickness tension stresses. Failure modes were specified with the user-defined results module in the following form:(3)$$\left\{ \begin{array}{l} \max\left[{\left(\frac{\sigma_{3c}}{F_{3c}}\right)}^{2}+{\left(\frac{E_{3}}{G_{13}}\right)}^{2}\cdot {\left(\frac{\tau_{13}}{F_{3c}}\right)}^{2};\;{\left(\frac{\tau_{13}}{F_{13}}\right)}^{2}+2\frac{G_{13}\cdot \sigma_{3c}}{E_{3}\cdot F_{13}}\right]=1,{\;\sigma }_{3}<0\\ \frac{\sigma_{3t}}{F_{3t}}=1,{\;\sigma }_{3}>0\; \end{array}\right.,$$
where σ_3*c*_ is through-thickness compression stresses, σ_3*t*_ is through-thickness tension stresses, τ_13_ is interlaminar shear stresses, *F*_3*c*_ is ultimate through-thickness compression strength, *F*_3*t*_ is ultimate through-thickness tensile strength, *F*_13_ is ultimate interlaminar shear strength, *E*_3_ и *G*_13_ are through-thickness modulus and interlaminar shear modulus, respectively.

### 2.4. Test Procedure

The tests were carried out on a universal testing machine INSTRON 5900R with 100 kN load cell. All tests were performed at a cross-head rate of 1 mm/min. Through-thickness compression and interlaminar shear stresses in the gauge section were determined by Equations (1) and (2). The DIC method was used to analyse the uniformity of the strain distribution in the gauge zone of the specimen and to quantify shear strains. DIC parameters are shown in Table 2. The full-field deformations of the specimens in the local coordinate system of the material are defined as follows:(4)$$\varepsilon_{3}=\varepsilon_{x}\sin^{2}\alpha +\varepsilon_{y}\cos^{2}\alpha -0.5\gamma_{xy}\sin2\alpha ,$$
(5)$$\gamma_{13}=-\varepsilon_{x}\sin2\alpha +\varepsilon_{y}\sin2\alpha +\gamma_{xy}\cos2\alpha .$$
*ε_3_*, *γ_13_* are through-thickness and interlaminar shear strains in the material axes, *ε_x_*, *ε_y_*, *γ_xy_* are normal and shear strains in the global coordinate system.

The specimen in a new fixture is shown in Figure 5a. Biaxial tests were performed by varying the loading angle in 5° steps. All specimens were loaded to failure. For each loading case, three specimens were tested. To construct the stress–strain curves, the engineering strain was determined by averaging the DIC strain measurements on the area with dimensions of 16 × 1 mm, which span the specimen gauge section (Figure 5b). The normalized strain was defined as the ratio of the local strain to the average strain over the gauge section.

## 3. Results and Discussion

### 3.1. Numerical Analysis

The influence of the specimen height on the uniformity of stress distribution in the gauge section was determined at the loading angle of 45°. This ensured the equality of the average through-thickness compressive and interlaminar shear stresses. The specimen heights of 15 mm and 20 mm, providing stable fixation, were considered. The stress distribution between notches at different specimen heights is shown in Figure 6. The choice of specimen height did not have a significant influence on the uniformity of the stress distribution in the gauge section. In both cases, uniform stress distribution was ensured in the central part of the gauge section. In the 15-mm specimen height, the average interlaminar shear and through-thickness compressive stresses in the gauge section of the specimen were 65.3 MPa and −64.4 MPa, respectively. With the increase in the specimen height to 20 mm, the average interlaminar shear and through-thickness compressive stresses remained at practically the same level and achieved 63.8 MPa and −63 MPa, respectively.

Then, the stress uniformity in the gauge section along the face and middle surfaces of the specimen was assessed. The stress distribution between the notches on the face (S1) and the middle surface (S2) of the specimen is shown in Figure 7. An almost identical stress distribution was provided on the face and middle surfaces of the specimen. On the face surface (S1), the average interlaminar shear and through-thickness compressive stress in the gauge section of the specimen were 63.8 MPa and −63 MPa, respectively. On the middle surface (S2) of the specimen, the interlaminar shear and through-thickness compressive stress in the gauge section were 62 MPa and −65 MPa, respectively. A slight difference in stress (less than 5%) confirmed that the stress distribution in the gauge section remained almost constant across the width. Thus, based on an estimate of the stress–strain state of the specimen face surface, the interlaminar parameters of composites under biaxial loading (45°) can be predicted.

A detailed stress analysis on the face surface of the specimen (S1) under different compression–shear loading was performed to determine the possible failure zone. An analysis of the stress fields at a loading angle α = 45° showed the localization of the maximum shear stresses in several areas (Figure 8a). The shear stress achieved 69 MPa in the gauge section of the specimen, while, near the notch flanks, the stress was approximately at the same level of 66 MPa. The maximum through-thickness compressive stresses were localized near the notch root and achieved 212 MPa (Figure 8b). Failure index maps of NU-Daniel theory for this combination of compression-shear loading also indicated two possible failure regions with almost the same failure index (Figure 8c). Since shear behaviour was considered linear in numerical analysis, it was not possible to draw a firm conclusion regarding whether premature failure of the specimen would occur near the notch flanks.

A significant difference in the stress distribution was observed at small loading angles (α < 20°). The maximum shear stresses at the loading angle α = 15° were localized near the notch flanks and achieved 83 MPa (Figure 9a). In this case, the shear stress in the gauge section for the corresponding combination of compression–shear loading was about 52 MPa. The maximum through-thickness compressive stresses were still localized at the notch root and achieved 542 MPa (Figure 9b). Failure index maps of the NU-Daniel theory for this combination of compression–shear loading testified the localization of failure near the notch flanks (Figure 9c).

Figure 10 shows the distribution of stresses and failure indexes along the gauge section. Sufficient uniform stress distribution was ensured in the central part of the gauge section. The combination of interlaminar shear and through-thickness compressive stresses in the gauge section led to the different distribution of failure indexes (Figure 9c). Delamination was predicted in the central region of the gauge section at the loading angle of 45°, while a significant safety margin was provided at the loading angle of 15°. It was previously shown that the possible failure zone was localized outside the gauge section of the specimen at the loading angle of 15° (Figure 9c). In this case, the interlaminar strength under combined compression-shear loading will be underestimated due to the premature failure of the specimen.

The combination of compression-shear loads corresponding to delamination in the gauge section was used as an initial approximation for the experiments.

### 3.2. Failure and Strain Distribution Analysis

During the visual inspection of the failed specimens, the failure regions that occurred under combined loading were identified (Figure 11). A mixed fracture mechanism near the notch flanks was observed at the loading angle α = 15°, while delamination was directly observed in the gauge zone at the loading angles α = 20–45°. Both cases of failure were qualitatively consistent with the results of preliminary numerical calculations.

According to the DIC measurements, an insignificant rotation of the specimens relative to the axis of symmetry during loading was established. This was about 1° at the loading angle α = 20°.

In a detailed analysis of the deformation fields, it was found that at the loading angles α < 35°, through-thickness strains were uniformly distributed over the gauge section in the stage of elastic deformation up to significant loads (Figure 12a for α = 20°). The distribution of through-thickness stresses looked similar in numerical calculations (see Figure 9b). At this stage, the uniform distribution of through-thickness stresses in the central part of the gauge section of the specimen was in a good agreement with the experimentally observed one. However, at the maximum load (before failure), a significant decrease in the through-thickness compressive strain in the central part of the gauge section of the specimen was observed due to damage accumulation (Figure 12b).

It should be noted that the shear strain distribution in the gauge section became less uniform with increasing loading angle (Figure 13c for α = 20° and 45°). However, the shear strain almost halved far from the gauge section (Figure 13a,b), indicating the direct localization of shear strains between notches. The distribution of interlaminar shear stresses in the numerical calculations differed due to the stress concentration near the notch flanks (see Figure 8a). A more uniform distribution of shear strains in the gauge section was observed in experiments at small loading angles (α < 20°). This is because the preliminary FEM is qualitative and simulations including shear nonlinearity would be needed to accurately determine the interlaminar shear strength of the specimens. It is worth noting that considering shear nonlinearity would result in a more uniform shear-stress state.

Figure 14 shows the distribution of through-thickness and interlaminar shear strains in the gauge section, obtained using the DIC method in the local coordinate system under combined loading. A sufficiently uniform distribution of through-thickness and shear strains in the gauge section was provided at loading angles α = 20–30°. The region of uniform distribution of deformations ranged from 0.15 to 0.85 of the normalized gauge section length. At the same time, the region of the deformations uniform distribution ranged from 0.3 to 0.7 of the normalized gauge section length with the increase in the loading angle from α to 45°. It should be noted that the average shear strains in all cases gave a lower estimate of the actual strains in the central part of the working section due to the stress concentration near the notches (Figure 14a). In this case, the actual interlaminar properties of the material are obviously underestimated.

### 3.3. Analysis of Stress–Strain Response

Typical stress–strain curves under biaxial loading are shown in Figure 15 and Figure 16. It was found that biaxial loading not only had a significant impact on the ultimate strength of the composite, but also on the load–displacement response. The initial slope in the curves, i.e., the apparent interlaminar shear modulus and the through-thickness modulus, did not significantly change under combined loading (Figure 15 and Figure 16). The average value (and RMS) of the apparent interlaminar shear modulus and the through-thickness modulus were 3.5 GPa and 13 Gpa, respectively. This estimate is in reasonable agreement with the known experimental data [33]. A non-linear and monotonically increasing behaviour was typical for shear stress–strain curves in all loading cases. A similar nonlinear behaviour was observed for shear stress–strain curves under biaxial loading in [8]. Two different sections of the compressive stress–strain curves were observed at α = 20–30°. In the first section, a monotonous increase in stresses was recorded up to the maximum compressive strain was recorded. In the second section, a sharp decrease in deformation occurred, with a further increase in stresses.

Thus, the maximum compression strain on the stress–strain curve corresponded to the moment at which composite structure discontinuity occurs. The disproportionate changes in through-thickness strains in the gauge section of the specimens were noted in these cases. After reaching the maximum compressive strain, the strain curves are likely to reflect the specimen behaviour rather than the material bevaviour.

### 3.4. Analysis of Strength Properties

The interlaminar shear strength under biaxial loading, determined at maximum load and at maximum compressive strain, was compared with the NU-Daniel theory. The elastic properties of the failure curve were previously determined from the stress–strain curves (Figure 15 and Figure 16). The interlaminar shear strength F_13_ = 48.4 MPa was determined by tests using the short beam method [40].

The average interlaminar shear and through-thickness compressive stresses at different loading angles are given in Table 3. The observed increase in the interlaminar shear strength under biaxial loading (by about two times) on the new fixture is in good agreement with the experimental data presented in [17]. At the same time, the experimental values of interlaminar shear strength are more repeatable in this study. The difference between the two methods of assessing the interlaminar shear strength significantly increased through-thickness compressive stresses in the gauge section. At the maximum value of transversal compressive stresses (α = 20°), this difference exceeded 20%. Figure 17 shows a comparison of the interlaminar shear strength, assessed by the maximum load and the maximum compressive strain with NU-Daniel theory.

Both methods of assessing the interlaminar strength under biaxial loading in the new fixture showed good agreement with the failure curve in most cases. It is expedient to focus on the stable behaviour of the material to avoid material discontinuities during the exploitation of actual composite elements. Otherwise, when the CFRP strength is evaluated according to the maximum load at the design stage, the risk of premature failure of the composite elements during exploitation will significantly increase.

## 4. Conclusions

In this article, a new experimental approach to studying the interlaminar shear properties of PMCs under biaxial loading was proposed. The approbation of the approach was carried out on the woven CFRP. During the experiments on the biaxial loading of composite specimens, the abilities of the new test fixture were demonstrated. The fixture could not only determine the strength properties of the material, it could also obtain stress–strain curves that can be used to develop and validate nonlinear numerical material models.

Among the most significant results, we can distinguish the following:The range of α = 20–45° corresponded to delamination in the gauge section of the specimen. The fracture mechanism changed at loading angles of 15° or less.At angles α < 35°, the through-thickness strains were uniformly distributed over the gauge section at the elastic deformation stage of the specimen. With further load increases, a significant decrease in the through-thickness compressive strain was observed in the central part of the gauge section before failure due to damage accumulation.The interlaminar shear modulus and the through-thickness compressive modulus insignificantly changed under different compression-shear loadings. The average values of these parameters were 3.5 GPa and 13 GPa, respectively.The value of the maximum compression strain on the stress–strain curve at α = 20–30° corresponded to the moment at which composite structure discontinuity occurred.With the increase in through-thickness compression stresses, the difference between the shear strength values, determined by the maximum load and the maximum compressive strain, increased up to 20% at the loading angle α = 20°.

In summary, it is important to note that a detailed analysis of both shear and through-thickness deformations is necessary to determine the moment at which material discontinuity occurs. In this case, stress–strain curves can be used in numerical material models to correctly assess the stress–strain state of composite structures.

## Figures and Tables

**Figure 1 polymers-14-02575-f001:**
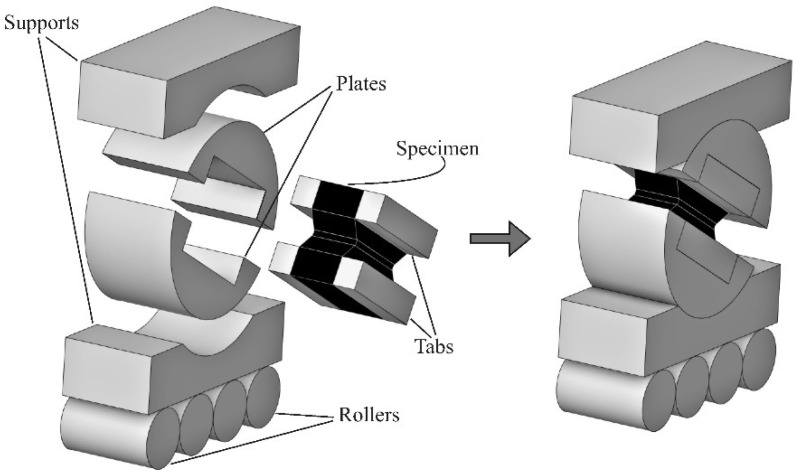
Three-dimensional visualisation of a new device to test PMCs specimens under biaxial loading.

**Figure 2 polymers-14-02575-f002:**
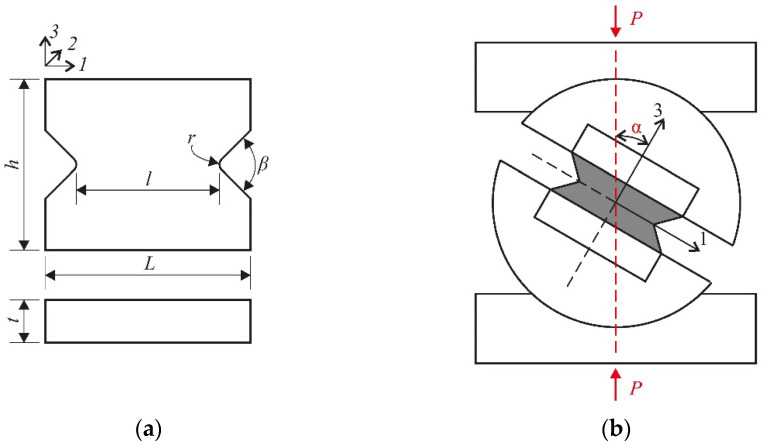
Dimensions of the specimen (**a**) and specimen-loading scheme (**b**).

**Figure 3 polymers-14-02575-f003:**
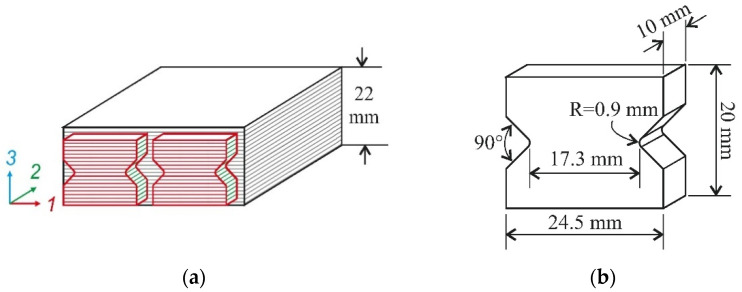
Cutting scheme (**a**) and dimensions of the specimens (**b**).

**Figure 4 polymers-14-02575-f004:**
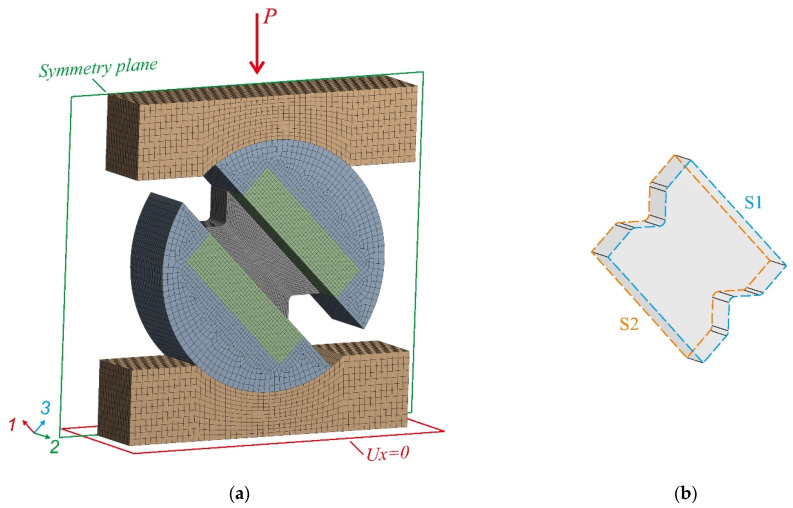
Finite-element mesh, boundary conditions, and loads (**a**); face (S1) and middle (S2) surfaces of the specimen (**b**).

**Figure 5 polymers-14-02575-f005:**
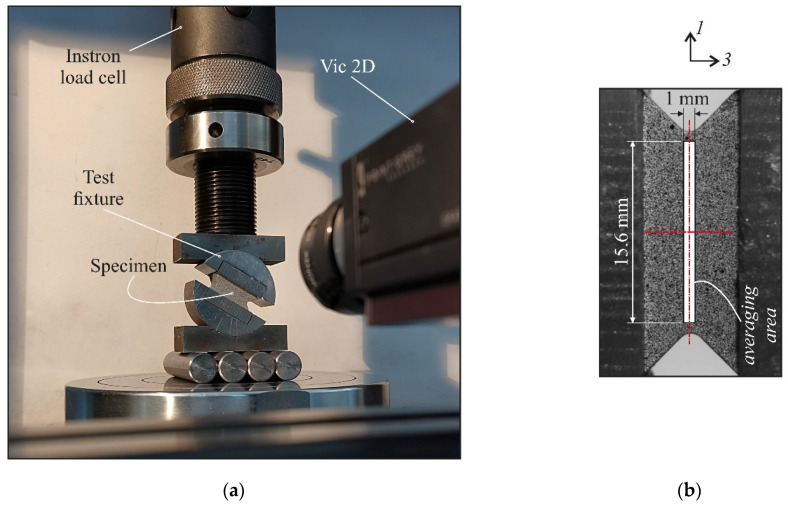
Experimental setup for testing under combined loading (**a**); specimen and strain averaging area (**b**).

**Figure 6 polymers-14-02575-f006:**
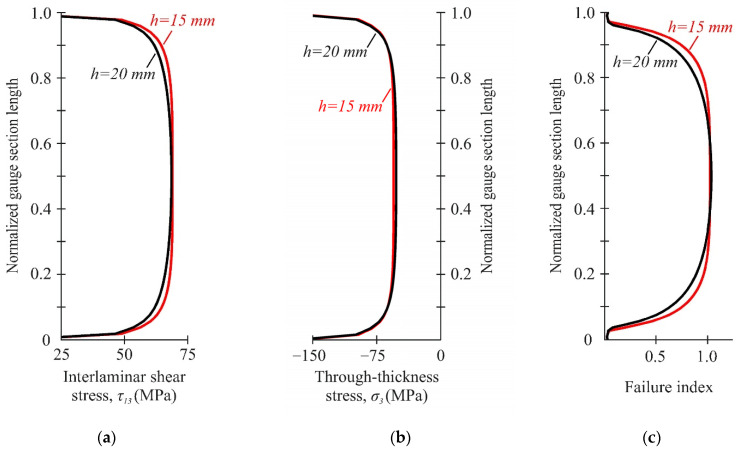
Distribution of interlaminar shear stresses (**a**), through-thickness stresses (**b**), and failure indexes NU-Daniel (**c**) between the notches at different specimen heights.

**Figure 7 polymers-14-02575-f007:**
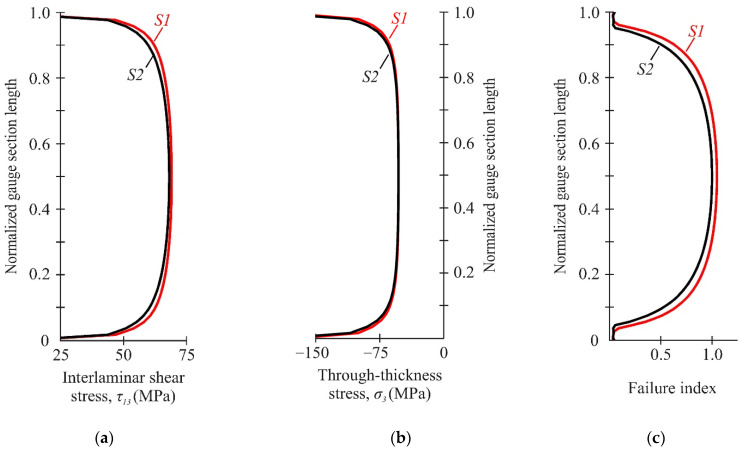
Distribution of interlaminar shear stresses (**a**), through-thickness stresses (**b**), and failure indexes NU-Daniel (**c**) between the notches on the face (S1) and the middle (S2) surfaces.

**Figure 8 polymers-14-02575-f008:**
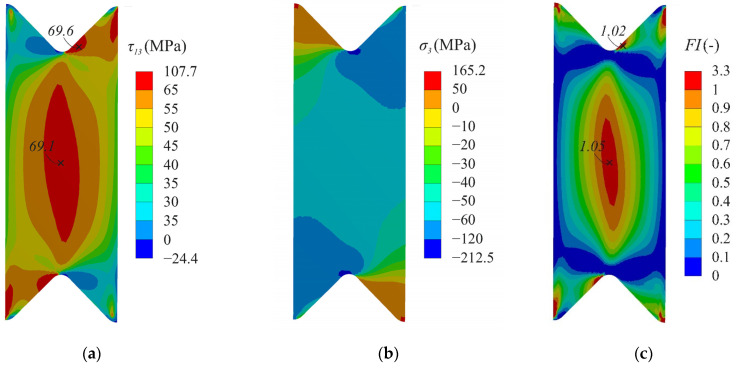
Distribution of interlaminar shear stresses (**a**), through-thickness stresses (**b**) and failure indexes NU-Daniel (**c**) on the face surface of the specimen at the loading angle α = 45°.

**Figure 9 polymers-14-02575-f009:**
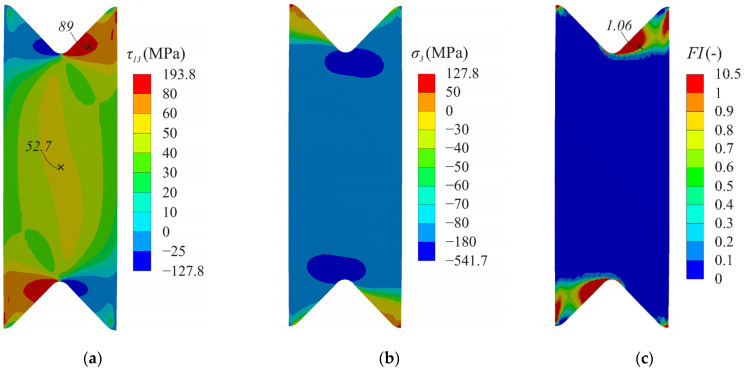
Distribution of interlaminar shear stresses (**a**), through-thickness stresses (**b**) and failure indexes NU-Daniel (**c**) on the face surface of the specimen at loading angle α = 15°.

**Figure 10 polymers-14-02575-f010:**
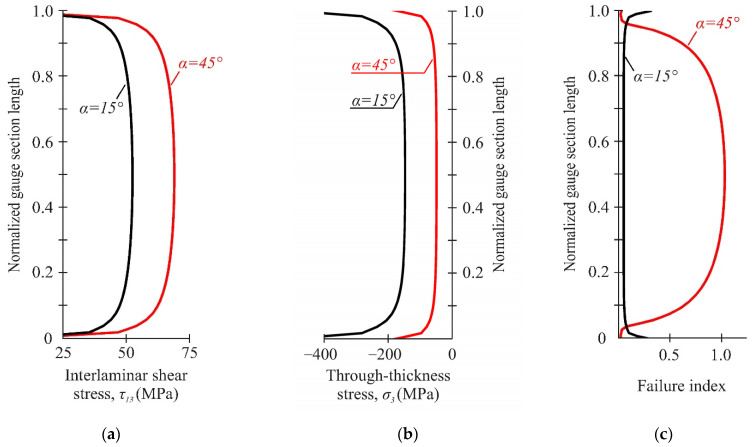
Distribution of interlaminar shear stresses (**a**), through-thickness stresses (**b**), and failure indexes NU-Daniel (**c**) between the notches at different loading angles.

**Figure 11 polymers-14-02575-f011:**
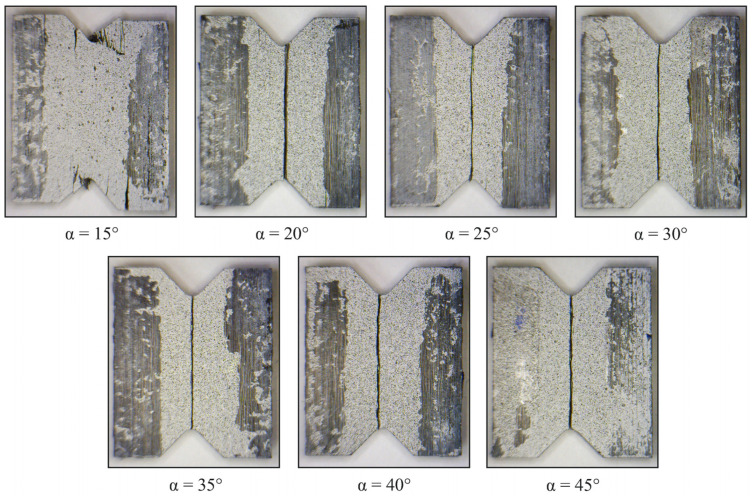
Failed specimens after combined compression/shear test.

**Figure 12 polymers-14-02575-f012:**
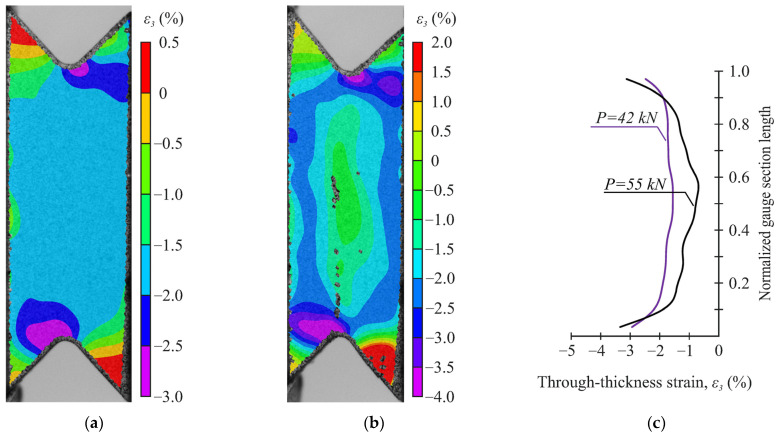
Distribution of through-thickness strains at α = 20°: (**a**) on the face surface of the specimen under load 42 kN, (**b**) on the face surface of the specimen under load 55 kN, (**c**) between the notches.

**Figure 13 polymers-14-02575-f013:**
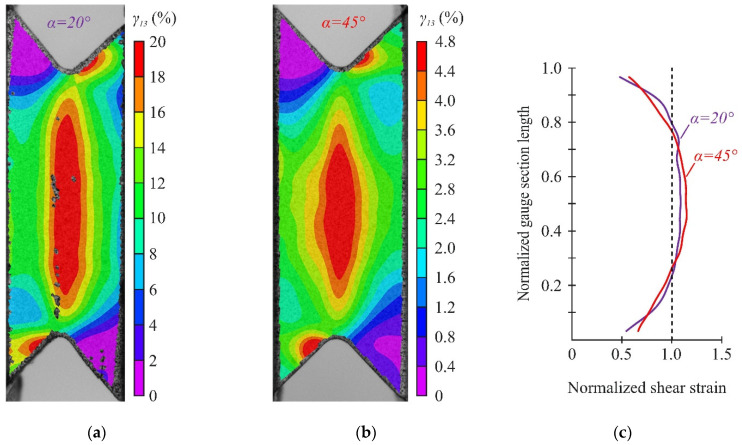
Distribution of shear strain at maximum load: (**a**) on the face surface at α = 20°, (**b**) on the face surface at α = 45°, (**c**) between the notches.

**Figure 14 polymers-14-02575-f014:**
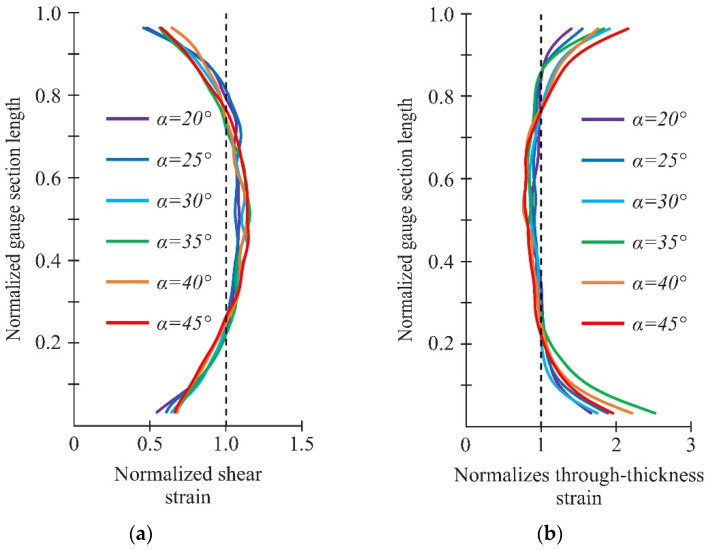
Distribution of interlaminar shear (**a**) and through-thickness (**b**) strains at the loading angles α = 20–45°.

**Figure 15 polymers-14-02575-f015:**
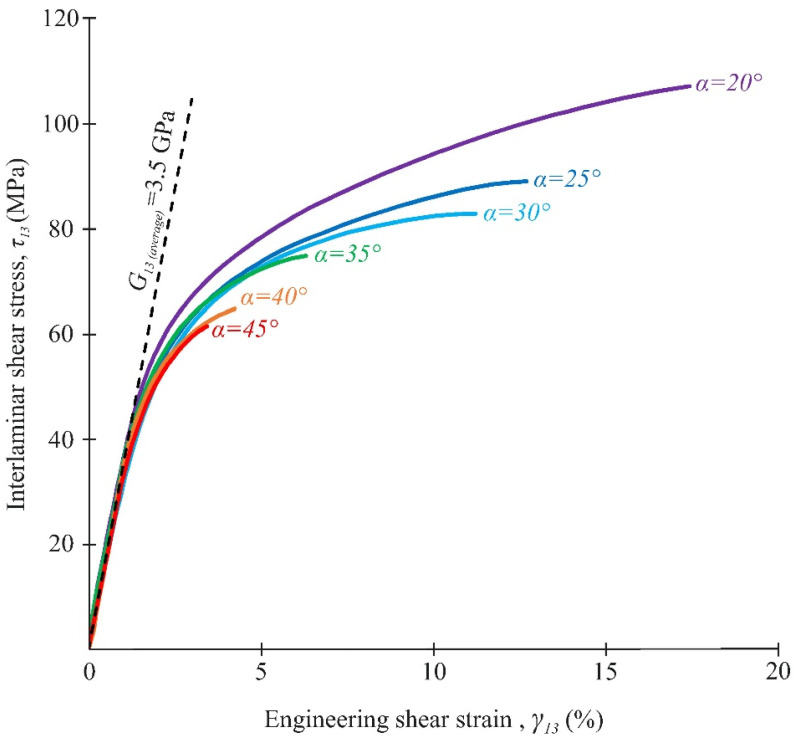
Shear stress–strain curves under combined compression/shear loading.

**Figure 16 polymers-14-02575-f016:**
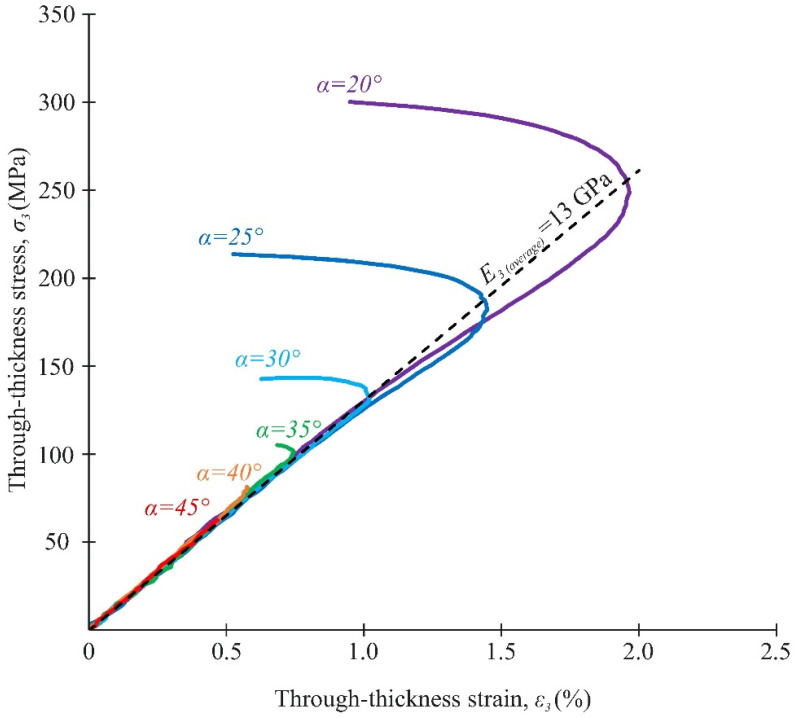
Through-thickness stress–strain curves under combined compression/shear loading.

**Figure 17 polymers-14-02575-f017:**
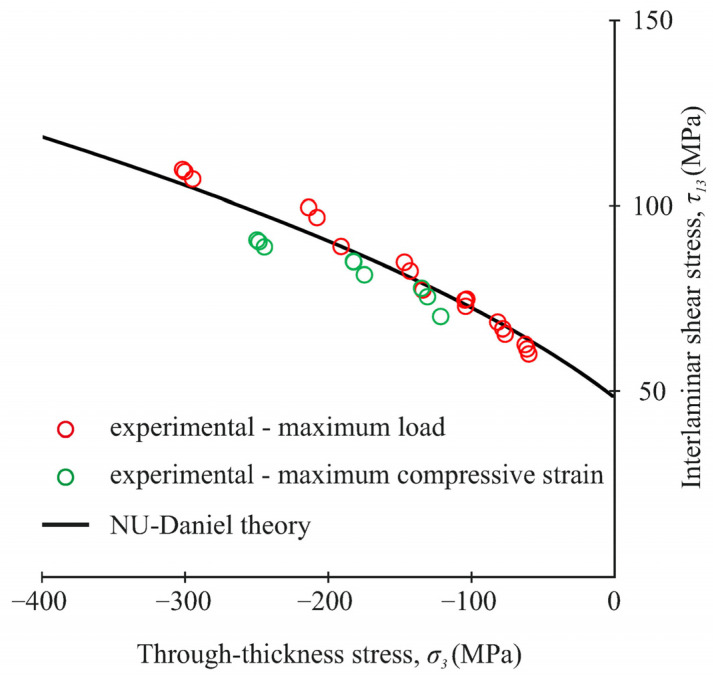
Comparison of the interlaminar shear strength, assessing with the maximum load and the maximum compressive strain with NU-Daniel theory.

**Table 1 polymers-14-02575-t001:** Elastic and strength properties of the CFRP.

Material Parameters	Value
Warp modulus *E*_1_ (GPa)	69.4 [26]
Weft modulus *E*_2_ (GPa)	69.4 [26]
Through-thickness modulus *E*_3_ (GPa)	11.5 [26]
In-plane shear modulus *G*_12_ (GPa)	5.0 [26]
Interlaminar shear moduli *G*_13_ *= G*_23_ (GPa)	3.4 [33]
Major Poisson’s ratio *ν*_12_	0.08 [26]
Through-thickness Poisson’s ratios *ν*_13_ *= ν*_23_	0.53 [26]
Through-thickness tensile strength *F*_3*t*_ (MPa)	63 [32]
Through-thickness compressive strength *F*_3*c*_ (MPa)	907 [26]
Interlaminar shear strength *F*_13_ (MPa)	54.3 [33]

**Table 2 polymers-14-02575-t002:** DIC parameters.

Camera	Point Grey GRAS 50S5M-C
Sensor	8-bit, 2048 × 2048 px
Lenses	Schneider Kreuznach Xenoplan 1.4/17
DIC Software	VIC 2D
Subset size	13 × 13 px

**Table 3 polymers-14-02575-t003:** Experimental assessment of stresses using two methods under biaxial loading.

Angle	Interlaminar Shear at Maximum Load	Interlaminar Shear at Maximum Compressive Strain	Through-Thickness Stresses at Maximum Load	Through-Thickness Stresses at Maximum Compressive Strain	Difference in Interlaminar Shear Stresses	Difference in Through-Thickness Stresses
20	109.02 ± 1.01	90.34 ± 0.73	299.52 ± 2.78	248.2 ± 2.0	20.67	20.67
25	95.48 ± 4.1	84.1 ± 1.57	204.78 ± 8.8	180.34 ± 3.4	13.47	13.49
30	81.84 ± 2.79	74.79 ± 2.89	141.74 ± 4.83	129.54 ± 5.01	9.45	9.45
35	74.73 ± 0.8		104.47 ± 0.45		-	-
40	67.27 ± 1.12		79.24 ± 1.95		-	-
45	61.7 ± 0.87		61.7 ± 0.87		-	-

## Data Availability

The data presented in this study are available on request from the corresponding authors.

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
