# Peer review of "Determination of Interlaminar Shear Properties of Fibre-Reinforced Composites under Biaxial Loading: A New Experimental Approach"

_polymers, 2022, doi:10.3390/polym14132575_

Round 1

Reviewer 1 Report

An innovative design was proposed to obtain the interlaminar shear properties of fibre-reinforced composites under biaxial loading. FEM and DIC analysis are used to obtain the full field deformation and failure mode of the composite material during the loading. Overall, this paper is well designed with some innovative ideas. However, the writing still needs further improvement. Please see the following specific comments.

1)   In the abstract part, please provide some application scenarios and research significance of biaxial tests for carbon/epoxy laminates. In addition, some quantitative results are preferable for readers to understand the current research work. How to determine the reliability of the new experimental approach for determination of interlaminar shear properties of laminates? Is it relatively compared with the existing methods?

2)  The introduction writing lacks some basic information about fiber reinforced polymer composites, especially carbon fiber reinforced composites (CFRP), such as composition, properties, advantage and type, etc, which can help readers better understand the polymer matrix composites. Please refer to the latest research below to fill the above gaps. International Journal of Fatigue, 2020, 134: 105480. Materials and Structures, 2020, 53: 73.

3)  DIC method has been proved to be a very effective method to detect the full-field deformation and strain of FRP during the loading, which is very important to reveal the failure mechanism of materials. In this paper, the authors have adopted the method to obtain full field deformation and failure mode. However, the summary about the monitoring of deformation or strain of composites through DIC technology should also be added to the introduction. Please refer to the following research on using DIC technology to monitor the strain distribution and failure mode of composites during the short beam shear loading: https://doi.org/10.1080/15376494.2021.1974620.

4)   The part 2 should be the material and method part, which should mainly include raw material, the design of devices and test methods. The current writing lacks necessary information and content. In addition, the biaxial loading test device in the current paper is independently designed by the authors, so other relevant reference summary is not appropriate here, they should be put in the introduction. In this part, the authors should state their own work.

5)   In Table 1, on the performance parameters of CFRP, why do you refer to different references to obtain them? Are these parameters consistent with the CFRP performance parameters used in this article?

6)     In part 3.2, reviewers did not see the accuracy verification of finite element model by the authors. Therefore, the reliability of some models and related analysis (Figure 4-8) are questionable.

7)     Some information about materials, sample and test procedure in part 4 should be put in part 2, and part 4 should only focus on the results and discussion of the experiment.

8)     Please add the comparative analysis between the test results of DIC and finite element simulation. This can effectively verify the accuracy of the finite element model.

9)    There are two “4.4 Analysis of stress-strain response”, Please check the repeatability and make any further modifications.

10) Please further refine the current conclusions, including 3~4 key points. In addition, please add some latest research in recent 2~3 years to the list of references.

Reviewer 2 Report

The paper is very interesting and good. The authors had made a significant contribution bu using a new experimental approach for determination of interlaminar shear properties of fiber reinforced composites under biaxial loading. Following are my minor suggestion.

1. Please modify the title as "An experimental and numerical study on interlaminar shear properties of fiber-reinforced composites".

2. Add more relevant keywords.

3. Too much paragraphs in introduction. Please reduce them.

4. Please write clearly the objective and significance of your work in the last paragraph of introduction.

5. Please improve all the Figure quality.

6. A comparison with previous study is necessary.

7. Add a discussion section before conclusion regarding practical implementation of current study.

8. Please make bullet point in conclusion.

9. What are your future recommendation.

10. Moderate English changes required.

Round 2

Reviewer 1 Report

Although the authors provided a cover letter document, the basis or explanation for some revision was not shown in detail in the cover letter. In addition, some key comments were not considered in the revised manuscript. The quality of papers has not been greatly improved.

Round 3

Reviewer 1 Report

The second point-to-point reply has greatly improved the quality of this paper. It is recommended to accept the current paper.